# Chemical Constitution and Antimicrobial Activity of Kefir Fermented Beverage

**DOI:** 10.3390/molecules26092635

**Published:** 2021-04-30

**Authors:** Abdul-Raouf Al-Mohammadi, Rehab A. Ibrahim, Ahmed H. Moustafa, Ahmed A. Ismaiel, Azza Abou Zeid, Gamal Enan

**Affiliations:** 1Department of Sciences, King Khalid Military Academy, Riyadh 11459, Saudi Arabia; almohammadi26@hotmail.com; 2Department of Botany and Microbiology, Faculty of Science, Zagazig University, Zagazig 44519, Egypt; rehabatef@yahoo.com (R.A.I.); ahmedismaiel@zu.edu.eg (A.A.I.); azza.abozeid@yahoo.com (A.A.Z.); 3Department of Chemistry, Faculty of Science, Zagazig University, Zagazig 44519, Egypt; ah_hu_mostafa@yahoo.com

**Keywords:** kefir beverage (KB), GC-MS analysis, pathogenic bacteria, fruit juices

## Abstract

Kefir beverage (KB) is a fermented milk initiated by kefir grains rich with starter probiotics. The KB produced in this study seemed to contain many chemical compounds elucidated by gas chromatography–mass spectrometry (GC-MS) and IR spectra. These compounds could be classified into different chemical groups such as alcohols, phenols, esters, fatty esters, unsaturated fatty esters, steroids, polyalkenes, heterocyclic compounds and aromatic aldehydes. Both KB and neutralized kefir beverage (NKB) inhibited some pathogenic bacteria including *Escherichia coli* ATCC11229 (*E. coli*), *Listeria monocytogenes* ATCC 4957 (*L. monocytogenes*), *Bacillus cereus* ATCC 14579 (*B. cereus*), *Salmonella typhimurium* ATCC 14028 (*Sal. typhimurium*) as well as some tested fungal strains such as *Aspergillus flavus* ATCC 16872 (*A. flavus*) and *Aspergillus niger* ATCC 20611 (*A. niger*), but the inhibitory activity of KB was more powerful than that obtained by NKB. It also appeared to contain four lactic acid bacteria species, one acetic acid bacterium and two yeast species. Finally, the KB inhibited distinctively both *S. aureus* and *Sal. typhimurium* bacteria in a brain heart infusion broth and in some Egyptian fruit juices, including those made with apples, guava, strawberries and tomatoes.

## 1. Introduction

Natural fermented foods are quite promising for the promotion of human health as they contain natural probiotics that improve many metabolic properties [1,2,3,4,5,6,7,8,9]. The metabolites of probiotics include enzymes that improve many human nutritional aspects. Other metabolites of probiotics such as organic acids, ethanol, acetaldehydes, and bacteriocins inhibit microbial pathogens [10,11,12]. These probiotic metabolites and other natural agents such as modified proteins of either plant or animal origin as well as plant extracts are quite promising to be used as food additives as they control food spoilage processes and could even kill the resistant variants of bacteria [13,14,15,16,17,18,19]. Thus, there is a need for further research on other fermented foods rich with probiotics.

Kefir is an acidic-alcoholic fermented milk beverage consumed all over the world that originated in the Balkans, Eastern Europe and the Caucasus [20,21,22,23]. Milk fermentation is traditionally achieved by inoculating milk with kefir grains, an example of symbiosis between yeast and bacteria held together by kefiran, a polysaccharide matrix [24,25]. Yeast and lactic acid bacteria coexisting symbiotically result in milk kefir fermentation [26].

Kefir grains are small, irregularly shaped, white to yellowish-white gelatinous masses varying in size from 0.3 to 3.5 cm in diameter; they are hard granules that resemble cauliflower blossoms. Their surfaces are adhered together by multiple biofilm products of probiotics to become granule-like three dimensional forms. The compositions of kefir grains differ from one country to another according to the liquid foods and starter cultures used for fermentation [27,28].

Several studies have demonstrated that the microorganisms present in kefir are probiotics of different bacterial species belonging to lactic or acetic acid bacteria such as *Latobacillus*, *Lactococcus*, *Leuconostoc*, *Pediococcus*, *Carnobacterium* and *Acetobacter* [29,30]. These probiotics have beneficial health properties, as they were reported to produce chemical metabolites with antibacterial [31], hypocholesterolemic [32], antihypertensive [33], anti-inflammatory [34], antioxidant [35] and anticarcinogenic effects [36]. Kefir’s antibacterial effect was recorded against many microbial pathogens [37].

Due to the importance of fermented foods for humanity since ancient times, fermentation is considered as one of the oldest food preservation methods, and fermented fruits and vegetables such as kefir juice beverages have been shown to have high nutritional value resulting from both their substrates, which have important antioxidant potential, and their cultures, which have proven probiotic properties [38]. Additionally, fermented fruit and vegetable juices are characterized by their acidic nature due to the presence of organic acids. They also contain many compounds with certain importance to humans, such as aldehydes, alcohols, heterocyclic compounds, steroids, polyalkenes, esters, phenols and fatty acids [39,40].

The antimicrobial activity of these fermented juices is due to the presence of probiotic metabolites such as antimicrobial proteins (bacteriocins), organic acids and diacetyls [41,42,43]. Kefir’s inhibitory activity has been reported to be effective against various species of pathogenic bacteria [43].

The WHO has recognized that kefir is a useful functional food for use as an alternative medicine and deemed research on it to be of particular interest [44,45].

The present study aims to (i) investigate kefir’s antimicrobial activity in vitro and in foods, (ii) isolate and identify kefir’s microbiota and (iii) determine kefir’s bioactive compounds by means of available instrumental analysis such as IR spectroscopy and GC-MS analysis.

## 2. Results

KB was prepared at our experimental conditions at an initial pH value of about 6.5, incubation temperature of 30 °C and incubation time of 24 h. The final pH of the KB prepared was 3.1 after 24 h of incubation and reached 2.6 after 48 h of incubation.

KB was subjected to GC-MS analysis to detect its bioactive compounds. The results given in Table 1 and Figure 1 show the compounds names and classes, in addition to molecular formula and molecular weight, for the chemical categories produced. The main compounds in the KB are alkaloids: 7-Tosyl-1,3:2,5:4,6-trimethylene-d-glycero-d-mannoheptitol; phenols: 2,2′-Methylenebis[6-tertbutyl]-*p*-cresol; esters: 2-Ethylhexyl phthalate, Phorobol 12,13-dihexanoate, 2,3-Dichloro 2-octyl phenyl fumarate, Nonyl octyl fumarate, 2-Chloro-6-(4-fluorophenyl)-2-octyl fumarate, 2-[(Methylsulfonyloxy)ethyl 4-(6-methyl 1,4-dioxaspiro [4.5]dec-7-yl) butanoate, cypermethrin, befinthrin, cyhalothrin, Dihydroobscurinervinediol diacetate, 3,4,5,6-Tetrahydro-6-nonul-2*H*-pyran-2-one (cyclic eb ster), 6-heptylotetrahydro-2*H*-pyran-2-one; fatty esters: Methyl hexadecanoate, Methyl octadec-16-enoate, Methyloctadec-10-enoate, 2-(Tetradecycloxyethyl) palmitate, Trimyristin, (E) -2(stearoyloxy) ethyl octadec-9-enoate; unsaturated fatty esters: Methyl 5,6-dihydro-5,6-dihydroxy-(5R, 6R)-10′-Apo-α′-PSI-carotenoate, Tetrahydrofurfuryl oleate, Methyl (10E) -12,12-dideutero-14-oxo-10-nonadecenoate; steroids: 17,17-Ethylenedioxy-5,19-cycloaandrast-6-en-3-one, (22E)-Ergosta-7,9 (11),22-trien-3-yl acetate, 28-Acetylspirosolan-3-yl acetate, 3-Oxo-9 *β*-lanosta-7-en-26,23-olide, Cholest-5-en-ol, 3-Methoxy-6-oxo-2′-methylenechloestano [7,8α] cyclobutane, 17-Acetoxy-4,4-dimethyl-3-methoxy-3,19-epoxy andorst-8-en-7-ol; polyalkene: 2,6,10,15,19,23-Hexa methyl-2,6,10,14,18,22-tetracosahexaene; heterocyclic compounds: 1-(2-Nitro-4-trifluoro-methyl-phenyl)-5-propyl-1*H*-[1,2,3] triazole-4-carboxylic ethyl ester, Isobutyl 6-methyl-2-oxo-4-[4-trifluoromethyl)phenyl]-1,2,3,4-tetrahydro-5-pyrimidinearboxylate; aromatic aldehydes: m-Phenoxy benzaldehyde. In addition, the IR spectrum (Figure 2) showed the presence of bands at 3450 cm^−1^ for OH, 2211 cm^−1^ for C≡N, 1738 cm^−1^ C=O for ester and cyclic ester, 1679 cm^−1^ C=O for amide and at 1589 cm^−1^ N=N for triazole. In addition, a band at 1325 cm^−1^ was characterized for the asym. SO_2_ group.

The produced KB was analyzed microbiologically. It was streaked onto the media specified in the Materials and Methods section. The isolated microorganisms could be classified into bacteria and yeasts based on the cultural characteristics of their colonies. All the microbial cultures were studied regarding cell morphology and Gram staining. Five bacterial cultures appeared. The microbial isolates were classified into four Gram positive rods and one Gram negative rod. The vegetative yeast cultures were Gram positive oval-shaped cells. API kits were used to identify all of the microbial species produced; one Gram negative bacterial strain was identified as belonging to *Acetobacter aceti*. However, the four Gram positive bacterial isolates (rod-cells) were identified as bacterial strains belonging to *Lactobacillus kefiranofaciens*, *Lactobacillus delbreuki* ssp. *bulgaricus*, *Lactobacillus acidophilus* and *Bifidobacterium bifidum*. The two yeast isolates were shown to be two strains belonging to *Saccharomyces cerevisiae* and *Saccharomyces turensis*.

The qualitative inhibition of some indicator microorganisms by either KB or NKB was studied. Results are given in Table 2. The KB inhibited both bacteria and fungi tested, and its inhibitory activity was distinctively more than that obtained by NKB. The inhibitory activity of both KB and NKB was more powerful against the bacteria tested than on the fungi tested.

Since the KB showed broader antibacterial activity than that obtained by NKB, it was used for the quantitative inhibition of the more sensitive bacterial strains; one Gram positive bacterial strain (*S. aureus*) and one Gram negative bacterial strain (*Sal. typhimurium*). Results on the inhibition of both *S. aureus* and *Sal. typhimurium* by KB in BHI broths are given in Figure 3A,B). Growth of the untreated cells of either *S. aureus* (Figure 3A) or *Sal. typhimurium* (Figure 3B) increased vigorously, reaching almost 7 log cycles increase within 72 h. However, growth of the treated cells of both two bacterial pathogens in BHI broths treated by either 2% or 4% *v*/*v* KB decreased distinctively (*p* value ≤ 0.05) and differences between values of log CFU/mL of controls and treated samples were almost 8 log cycles after 24, 48 h in all treatments. No growth of *S. aureus*; *Sal. typhimurium* in BHI broths treated with 4% KB was detected after 24 and 48 h of incubation respectively. Growth of both bacterial pathogens was not detected after 72 h of incubation in BHI broths treated by 2% KB (Figure 3A,B).

Sterile apple juice was treated with either 2% or 4% *v*/*v* KB and inoculated with 5.7 × 10^2^ CFU/mL of either *S*. *aureus* or *S. typhimurium*. Growth of the control cells of either *S*. *aureus* (Figure 4A) or *Sal. typhimurium* (Figure 4B) increased vigorously by almost 5 log cycles increase within 96 h. However, growth (CFU/mL) of the treated cells decreased distinctively (*p* value ≤ 0.05) and no growth of both bacterial pathogens was detected at the end of 96 h of incubation in apple juice treated by 2% KB. Cells treated by 4% KB showed no growth of *S. aureus*; *Sal. typhimurium* after 48 and 72 h, respectively (Figure 4A,B).

The inhibition of both *S. aureus* and *Sal. typhimurium* by KB (2% and 4%) in fresh guava juice was studied (Figure 5A,B). The untreated control cells increased by almost 5 log cycles within 96 h, but growth (CFU/mL) of the treated cells decreased distinctively (*p* value ≤ 0.5) and no growth of *S. aureus*; *Sal. typhimurium* was recorded after 48 h; 72 h in samples of guava juices treated by 4% KB; at the end of 96 h; 96 h in samples treated by 2% KB, respectively (Figure 5A,B).

Sterile strawberry juice was treated with either 2% or 4% *v*/*v* KB and inoculated with 5.7 × 10^2^ CFU/mL of either *S. aureus* or *Sal. typhimurium*. Results are given in Figure 6A,B. Growth of control cells increased, reaching almost 5.8–6.7; 8.3 × 10^7^ CFU/mL within 72–96 h for both organisms, but the growth of both pathogens decreased distinctively (*p* value ≤ 0.05) reaching zero after 48 h of incubation in samples treated by 4% KB. In strawberry juice samples treated by 2% KB, no growth of *S. aureus*; *Sal. typhimurium* was detected after 72 h; 96 h of incubation (Figure 6A,B).

The inhibition of both *S. aureus* and *Sal. typhimurium* in tomato juice by KB (2% and 4%) was studied. Results are given in Figure 7A,B. The untreated control cells increased by almost 5 log cycles within 96 h, but growth (CFU/mL) of the treated cells decreased distinctively (*p* value ≤ 0.05), and no growth of both *S. aureus* and *Sal. typhimurium* was recorded after 48 h; 72 h in tomato juice treated by 4%; 2% KB, respectively (Figure 7A,B).

## 3. Discussion

Studies on the chemical and microbiological composition of kefir are needed since kefir is rich in probiotics and prebiotics which have many nutritional benefits for humans; kefir probiotics produce metabolites such as enzymes, antioxidants, vitamins, and antimicrobial agents [46]. The kefir produced in this study is traditionally made by fermentation of cow’s milk with kefir grains (Egyptian made) [47]. Either kefir grains or KB differ among countries as the milk used is either cow’s milk, goat milk, sheep milk, or camel milk [48]. Kefir grains can also ferment soy milk, rice milk, nut milk, coconut milk, and fruit juices [49].

As a result of fermentation, the KB could contain many fermentation end products. Nine chemical groups were detected; all of them were reported to inhibit bacterial pathogens by different mechanisms of action [2].

Since alkaloids cause membrane damage and rapid denaturation of proteins, as well as nutrient leakage from the cell [50], they are to be antibacterial. This results in a defect in cell metabolism and cell lysis [51].

Phenols elucidated in this study appeared also in a previous study to induce antibacterial activity through progressive leakage of intracellular constituents, including K+. The first index of membrane disintegration [52], also inhibiting the uptake of essential nutrients, resulting in cell death.

Esters and fatty acid esters appeared herein are, in general, positively charged and more hydrophobic; such hydrophobicity allows electrostatic interactions with the bacterial cellular components, leading to loss of cell viability due to the formation of fully de-energized killed cells. They also act as surfactants which cause the inhibition of five foodborne pathogens, namely *B. cereus*, *B. subtilis*, *S. aureus*, *E. coli* O157: H7, and *Salmonella typhimurium*, and also act as antibacterial food additives through the prevention of bacterial growth and biofilm formation [53].

Steroids (cholesterol and ergosterol) are components of cell membranes. They have antimicrobial characteristics and are used to treat infections caused by Gram-negative and Gram-positive bacteria throughout the prevention of the normal development of the cell membrane, and also the disruption of cell integrity and permeability [54].

Polyalkenes that appeared herein also exert antimicrobial action due to the repulsive force formed between the bacteria (negatively charged) and the polymers (positively charged) which in turn causes inhibition of cell permeability [55].

Heterocyclic compounds have been reported to be used as an analgesic, anthelmintic, antitubercular, plant growth regulator, antiviral, antifungal, and anticancer agent [56]. They showed their antibacterial activity through their ability to interact with either electrophiles or nucleophiles of the cells, leading to the inhibition of cell wall synthesis, inhibition of protein synthesis, inhibition of DNA synthesis, inhibition of metabolic pathways, and interference with cell membrane integrity [57].

Finally, aromatic aldehydes elucidated in this study appeared also in a previous study to possess high bactericidal activity through the association with the outer layer of bacterial cells [58], specifically with unprotonated amines on the cell surface which in turn affect the transport of ions across the cell wall and on enzyme systems where access of substrate to an enzyme is prohibited [59]. It will be necessary to test the antimicrobial activity of each compound alone. 

The antibacterial activity showed herein by kefir compounds might also be due to osmotic pressure of the solutes which existed in the hypertonic medium with the outer aquatic medium; this facilitates the diffusion of the bioactive materials from cell membranes across the selective permeability. The lipophilic nature of some solutes facilitates their attachment to bacterial cell membranes which in turn causes cell death [18,19].

As the KB used herein is traditionally made in Egypt, it was necessary to isolate and identify its microbiota. The microbiota showed from KB herein in this study almost concur with other published studies [60]. Probiotic bacteria found in kefir include *Lactobacillus acidophilus*, *Bifidobacterium bifidum*, *Streptococcus thermophiles*, *Lactobacillus helveticus*, *Lact. Kefirofaciens, Lactococcus lactis*, *Leuconostoc* spp. and *Lactobacillus delbreukii* [61]. In addition to bacteria, kefir contains many yeast species such as *Saccharomyces cerevisiae*, *Saccharomyces turensis*, *Saccharomyces fragilis*, *Candida kefyr*, *Kluveromyces marixians* [62]. Certain kefir types do not need to contain all the above microorganisms, but the kefir content of microbiota differs from one type to another and this depends on many factors such as fermentation liquid, kefir grains used, sterilization conditions [63].

The antimicrobial potential of KB was more pronounced against the indicator bacteria used than fungi. This has coincided with the latter published work in this respect [64]. Organic acids produced decreased pH to final levels and it was around 2.5–3.0 herein which can inhibit both bacterial and fungal growth, but other metabolites of bacterial probiotics such as antimicrobial proteins (bacteriocins) and polyalkenes can inhibit bacteria but not fungi [1,9,17,65].

The obtained KB herein showed higher antimicrobial activity than the NKB. This result indicated that the inhibitory activity was not due to acidic pH only, but also due to many metabolites detected in KB in this study [66]. Withuhn et al. [67] reported that the antimicrobial activity of kefir beverage is not simply due to the low pH value but due to the consistency of specific inhibitory substances that react with each other in a specific manner. The antimicrobial activity of neutralized kefir suspension might be related to the presence of one or more of the detected kefir compounds in this study which interact with each other to enhance or antagonize their antimicrobial effects [68].

Since *S. aureus* and *Sal. typhimurium* were the more sensitive organisms to the obtained KB, they were used as indicator organisms in further experiments. In BHI broth and juices such as apple juice, guava juice, strawberry juice, and tomato juice, the obtained KB inhibited both *S. aureus* and *Sal. typhimurium*. This is a promising result for using the KB either as a juice additive or to be used as a starter and protective syrup during either vegetable or fruit juice fermentation [39,69]. Hence, there is a need to develop easy ways that can help in the reduction of foodborne bacterial pathogens in fresh juices by using KB as fruit and vegetable juices that are subjected to fast deterioration. The use of KB as a juice additive can extend the shelf-life of such juices [70].

From all the previous investigations, it can be concluded that kefir beverage is a promising food additive in preserving fresh juices and other food products. Silva et al. [71] reported that soymilk fermentation with kefir can greatly enhance the health-promoting properties of soymilk as the addition of kefir greatly increased the count of lactobacilli and conversely lowered lipid, ash, total solid, and carbohydrate contents. It also decreased caloric value and titratable acidity upon treatments with higher soymilk Kefir percentages.

Finally, KB could be used as an additive for either vegetable or fruit juices in Egypt to protect such juices which can be left outside refrigerators for about 12 h, However, further work will be necessary to inhibit other pathogenic bacteria by KB as an additive (sterilized KB) or during juice fermentation by native KB as starter cultures at storage conditions, and to investigate the antimicrobial activity from each kefir compound alone; work in this regard is in progress.

## 4. Materials and Methods

### 4.1. Kefir Grains

Kefir grains were provided from a local market in Sharkia Governorate (80 km north of Cairo), Egypt, as traditionally Egyptian made. They were round-shaped like cauliflower grains with a white to creamy yellow color. They were initially created by auto-aggregations of different lactic acid bacteria, acetic acid bacteria, and yeasts which produce polysaccharide biofilms, leading to the formation of adhered grain surfaces after successive fermentations of animal or plant milk (e.g., soy milk) [72].

### 4.2. Preparation of Kefir Beverage (KB)

For the preparation of kefir beverage, a total of 100 g of the kefir grains (starter inoculum) were inoculated into 1000 mL skimmed cow’s milk (10% *w*/*v*) and incubated at 25 °C for 24 h. At the end of the fermentation process, the grains and milk were separated using a sterilized cheesecloth filter (2 mm pore size) [73]. The obtained KB was subjected to both chemical and microbiological analysis immediately after its preparation. The KB aliquots used for inhibition of pathogenic bacteria were kept in the refrigerator at 4 °C until used (1–12 h).

### 4.3. Microbial Test Strains

The indicator bacterial strains used included both bacterial and fungal pathogens. The bacterial strains used included Gram-positive bacteria such as *Staphylococcus aureus* ATCC6538 (*S. aureus*), *Bacillus cereus* ATCC14579 (*B. cereus*), *Listeria monocytogenes* ATCC4957 (*L. monocytogenes*), and Gram-negative bacteria such as *Escherichia coli* ATCC11229 (*E. coli*) and *Salmonella typhimurium* ATCC14028 (*Sal. typhimurium*). These bacterial test strains were maintained in glass beads, stored at −20 °C, and subcultured into brain heart infusion broth (BHI broth, Oxoid).

The fungal test strains used included *Aspergillus flavus* ATCC16872 (*A. flavus*) and *Aspergillus niger* ATCC20611 (*A. niger*). These fungal cultures were maintained in glass beads, stored at −20 °C and subcultured into potato dextrose broth (Difco, Sparks, NV, USA).

### 4.4. Instrumental Analysis of KB

For analysis of the KB existing compounds, about 1 mL sample of the KB was added to a 20 mL screw-cap solid-phase microextraction (SPME) vial with a silicone/polytetrafluoroethylene septum (Apex Scientific, Maynooth, Ireland) and equilibrated to 75 °C for 5 min with pulsed agitation for 5 s at 400 rpm with a GC Sampler 80 (Agilent Technologies Ltd., Little Island, Cork, Ireland). A single 50/30-mcarboxen-divinylbenzene-poly dimethyl siloxane SPME fiber (Agilent Technologies Ltd., Ireland) was used. It was exposed to the headspace above the samples for 20 min at a depth of 1 cm at 75 °C. The fiber was retracted and injected into the GC inlet and desorbed for 2 min at 250 °C. After injection, the fiber was heated in a bake outstation for 3 min at 270 °C to clean the fiber. The samples were analyzed in triplicate. Injections were made on an Agilent 7890A GC apparatus with an Agilent DB-5 column (60 m by 0.25 mm by 0.25 m) with a multipurpose injector with a Merlin Microseal (Agilent Technologies Ltd., Ireland). The temperature of the column oven was set at 35 °C, held for 0.5 min, increased at 6.5 °C·min 1 to 230 °C, and then increased at 15 °C·min 1 to 325 °C, yielding a total run time of 36.8 min. The carrier gas was helium held at a constant pressure of 231b/in2. The detector was an Agilent 5975C MSD single-quadrupole mass spectrometer detector (Agilent Technologies Ltd., Ireland). The ion source temperature was 230 °C, the interface temperature was set at 280 °C, and the MS mode was electronic ionization (70 V) with the mass range scanned between 35 and 250 atomic mass units. Compounds were identified by comparison of their retention times and mass spectra with those of WILEY 09 and the National Institute of Standards and Technology 2011 mass spectral library (NIST 11) [74]. The automated mass spectral deconvolution and identification system, as well as an in-house library with target and qualifier ions and linear retention indices for each compound, were created in Target View software (Markes International Ltd-Llantrisant, United Kingdom). Auto tuning of the GC-MS system was carried out before the analysis to ensure optimal GC-MS performance [75]. 

Infrared spectra of the obtained KB were measured with a Fourier transform infrared (FTIR) spectrometer (Bruker Optik GmbH, Ettlingen, Germany) according to the method reported by Shang, Xu, and Li [76], to determine the presence of various functional groups in the obtained KB. The pellets for FTIR analysis were obtained by grinding a mixture of 1 mg of freeze-dried KB powder with 100 mg of dry potassium bromide powder (KBr), followed by pressing the mixture in a mold. The FT-IR spectra were recorded in the region of 4000–400 cm^−1^ at a resolution of 4 cm^−1^. The resulting data were processed using OPUS/IR NT4.0 spectroscopic software package (Bruker Optik GmbH) installed on the FTIR instrumentation.

### 4.5. Isolation and Characterization of Probiotic Microorganisms from the KB

Serial two-fold dilutions of the KB were made; then, 0.1 mL aliquots from these dilutions were pipetted onto MRS agar plates [77]; specific *Acetobacter* agar (Oxoid, UK); Sabaraoud agar (Oxoid) for isolation of lactic acid bacteria; acetic acid bacteria; yeasts, respectively. The agar plates were incubated at 35 °C for either 48 h for bacteria or 4 days for yeasts. Pure and single colonies of the obtained microbes were picked up by sterile needles and inoculated into Brain Heart Infusion broths (BHI broth, Oxoid). After 24 h of incubation at 35 °C, 100 µL aliquots of microbial suspensions were loaded aseptically onto API kits (BioMérieux, France) that were then used for identification of the microorganisms isolated as given by the manufacturer’s instructions. Both bacteria and fungi isolated were checked for their Gram stain and cell morphology using a light microscope [2,3,78,79].

### 4.6. Preparation of Fruit Juices

Fresh fruits of apple (*Malus domestica*), guava (*Psidium guajava*), strawberry (*Fragaria ananassa*) and tomato (*Solanum lycopersicum*) were purchased from local markets (Zagazig City, Sharkia Governorate, Egypt).

Fresh fruits were washed with sterile distilled water. One hundred grams of each fruit sample were mixed with sterile distilled water at the ratio of 1:1 (*w*/*v*) as described previously [80], then homogenized by using a mixer (Braun combimax 700 vital, Germany). The obtained fresh juice was then centrifuged at 15.000 rpm for 30 min at room temperature. The supernatant of each fresh juice was collected in glass bottles, sterilized by autoclaving at 120 °C for 15 min and was then used immediately. The remaining aliquots were kept in a refrigerator at 4 °C for 48 h.

### 4.7. Bioassay of the Kefir Beverage (KB)

The KB was prepared as described above. It was also neutralized (NKB) using 0.1N NaOH at pH 7.0. Both the KB and NKB were sterilized by Millipore filtration (0.45 µm, Amicon). The antimicrobial activity of both KB and NKB was studied using an agar well diffusion assay [81]. Brain Heart infusion agar plates (BHI agar, Oxoid) were prepared and inoculated by 5.7 × 10^2^ CFU/mL of the indicator bacteria. Additionally, potato dextrose agar plates (Oxoid) were prepared and inoculated by 10^5^ spores/mL of the indicator fungal strains used. Microbial inocula were spread onto the agar plates by sterile glass rods under completely aseptic conditions. Sterile syrigs were used to create wells (5 mm in diameter). Then, aliquots (0.1 mL) of either the KB or NKB were pipetted into the wells. The inoculated and treated agar plates were incubated at 35 °C for 48 h; 4–7 days for bacteria and fungi, respectively. Diameters of inhibition zones were calculated after 48 h; 4–7 days for the indicator bacteria; fungi, respectively, according to Clinical and Laboratory Standards Institute (CLSI) [82].

### 4.8. Inhibition of Both S. aureus and Sal. typhimurium in BHI Broth and Juices of Apple, Guava, Strawberry and Tomato

A series of 250 mL Erlenmeyer flasks (Gomhuria Company, Cairo, Egypt), each containing 100 mL aliquots of either BHI broth (Oxoid) or fruit juices, were sterilized by autoclaving at 120 °C for 15 min. After cooling, they were inoculated by 5.7 × 10^2^ CFU/mL of the indicator bacteria, treated by either 2% or 4% KB, and were then incubated in an incubator (New Brunswick Scien. Co, New Jersey, NJ, USA) at 3 °C for 4 days. Every 24 h, samples were withdrawn and the growth of the indicator bacteria (CFU/mL) was calculated by [81].

### 4.9. Statistical Analysis 

Results were expressed as the mean ± standard deviation (SD). Statistical significance was evaluated using analysis of variance (ANOVA) test (SAS version 9.1, SAS Institute, Inc., Cary, NC, USA) [83] followed by the least significant difference (LSD) test at 0.05 level. *p* value < 0.05 means significant but *p* value > 0.05 means nonsignificant [84].

## 5. Conclusions

The KB used in this study was chemically analyzed using IR spectrometry and GC-MS analysis. It was shown to contain many beneficial chemical compounds. It contained five probiotics. This KB inhibited both pathogenic bacteria and fungi. The inhibitory activity was more powerful against bacteria than fungi. 

## Figures and Tables

**Figure 1 molecules-26-02635-f001:**
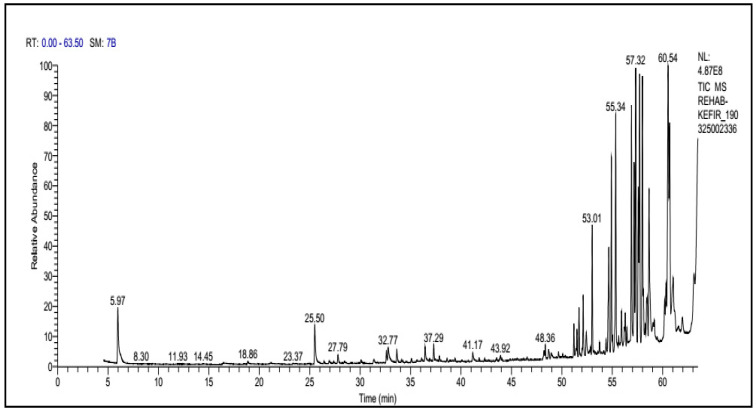
GC-MS analysis of KB.

**Figure 2 molecules-26-02635-f002:**
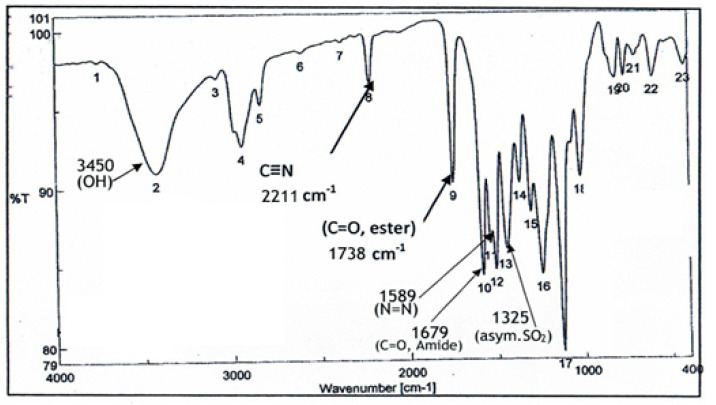
IR spectrum in KBr (discs) for the extraction of KB.

**Figure 3 molecules-26-02635-f003:**
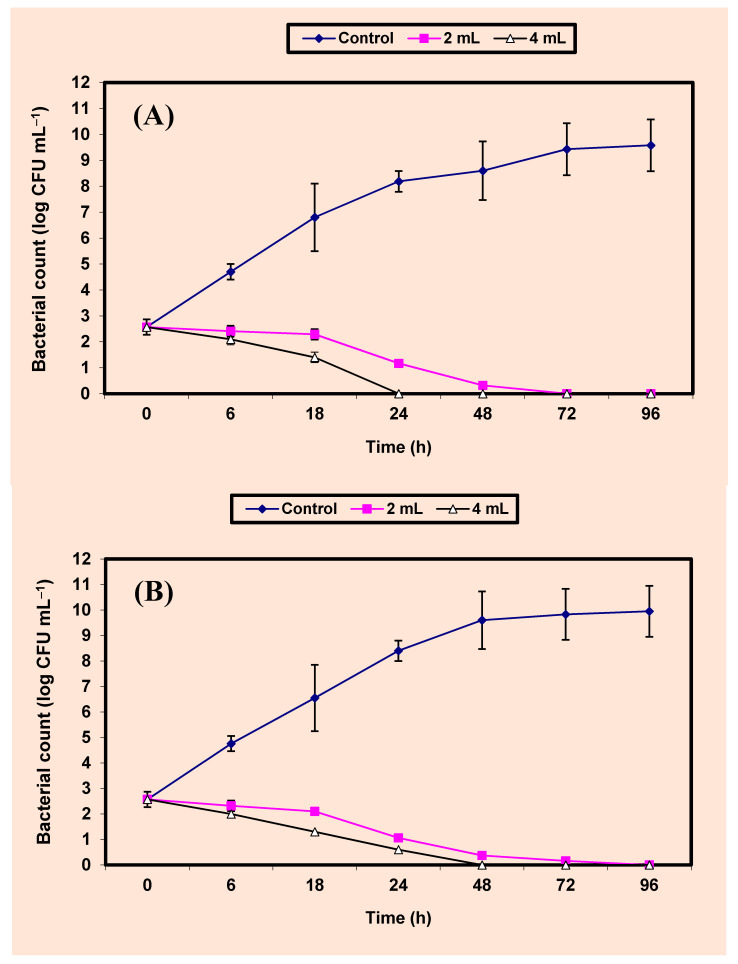
Inhibition of both *S. aureus* (**A**) and *Sal. typhimurium* (**B**) in BHI broth. Symbols ♦, ■, ∆ refer to control untreated samples, samples treated by 2%, 4% KB, respectively.

**Figure 4 molecules-26-02635-f004:**
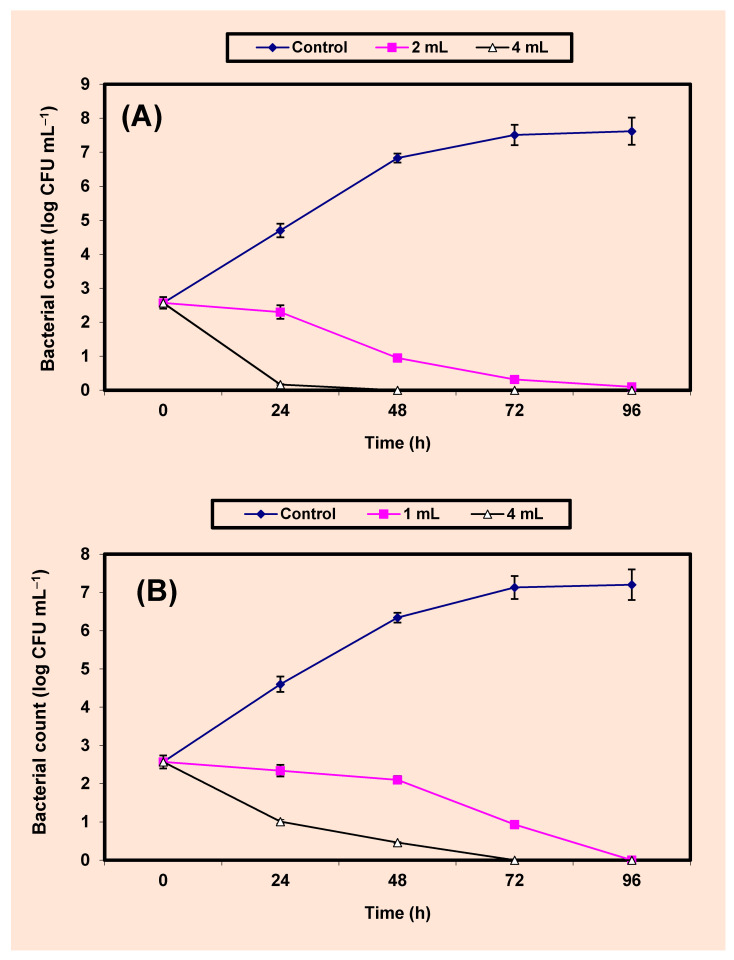
Inhibition of both *S. aureus* (**A**) and *Sal. typhimurium* (**B**) in apple juice. Symbols ♦, ■, ∆, refer to control untreated samples, samples treated with 2%, 4% KB, respectively.

**Figure 5 molecules-26-02635-f005:**
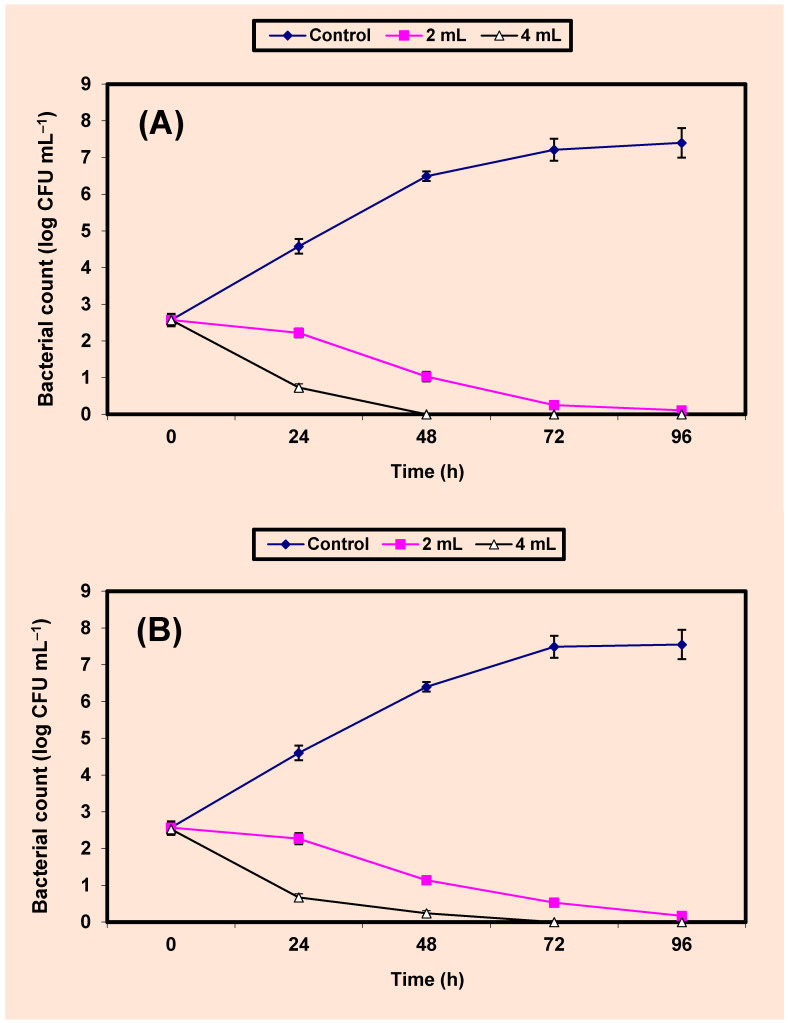
Inhibition of both *S. aureus* (**A**) and *Sal. typhimurium* (**B**) in guava juice. Symbols ♦, ■, ∆, refer to control untreated samples, samples treated with 2%, 4% KB, respectively.

**Figure 6 molecules-26-02635-f006:**
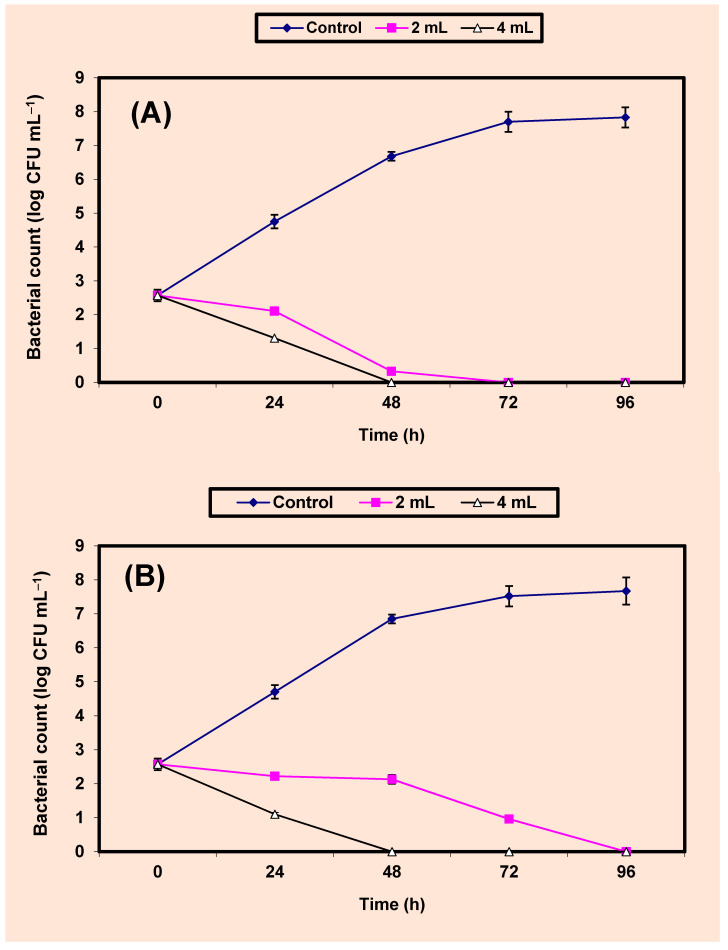
Inhibition of both *S. aureus* (**A**) and *Sal. typhimurium* (**B**) in strawberry juice. Symbols ♦, ■, ∆, refer to control untreated samples, juice samples treated with 2%, 4% KB.

**Figure 7 molecules-26-02635-f007:**
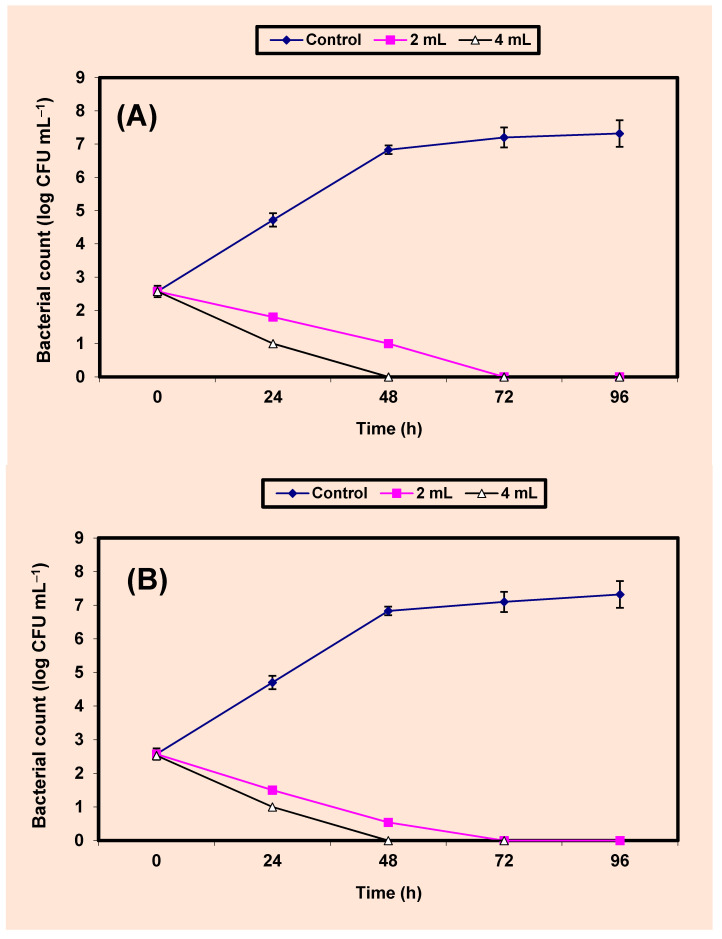
Inhibition of both *S. aureus* (**A**) and *Sal. typhimurium* (**B**) in tomato juice. Symbols ♦, ■, ∆, refer to control untreated samples, samples treated with 2%, 4% KB, respectively.

**Table 1 molecules-26-02635-t001:** Putative identification of the chemical components from KB when subjected to GC-MS (gas liquid chromatographic–mass spectrometry).

No.	Classification, Compound Name and Structure	Mol. Formula and Mol. Wt.	Area	Parent Ion (M^+^)	Base Peak (m/z) (100%)
	**Group 1 (Alkaloids)**				
**1**	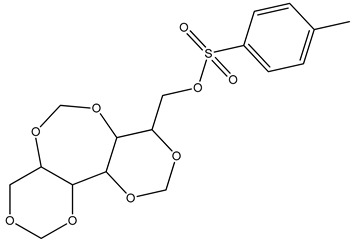 7-Tosyl-1,3:2,5:4,6-trimethylene-d-glycero-d-mannoheptitol	C_17_H_22_O_9_S (402.0)	0.77	402.0	91.00 and 155.0
	**Group 2 (Phenols)**				
**1**	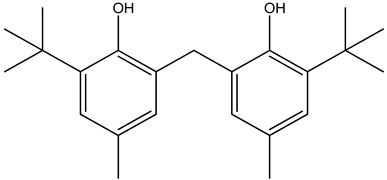 2,2′-Methylenebis[6-tertbutyl]-*p*-cresol	C_23_H_32_O_2_ (340.0)	0.28	340.0	177.0
	**Group 3 (Esters)**				
**1**	2-Ethylhexyl phthalate	C_24_H_38_O_4_ (390.0)	0.32	390.0 (M^+1^)	149.0
**2**	Phorobol 12,13-dihexanoate	C_32_H_48_O_8_ (560.0)	0.77	560.0	43.00
**3**	2,3-Dichloro 2-octyl phenyl fumarate	C_18_H_22_Cl_2_O_4_ (372.0)	5.13	372.0	99.0
**4**	Nonyl octyl fumarate	C_21_H_38_O_4_ (354.0)	5.13	355.0 (M^+1^)	71.00
**5**	2-Chloro-6-(4-fluorophenyl)-2-octyl fumarate	C_18_H_22_ClFO_4_ (356.0)	5.13	356.0	99.0
**6**	2-[(Methylsulfonyloxy)ethyl 4-(6-methyl 1,4-dioxaspiro [4.5]dec-7-yl)butanoate	C_16_H_22_O_7_S (364.0)	0.42	364.0	99.00 and 111.0
**7**	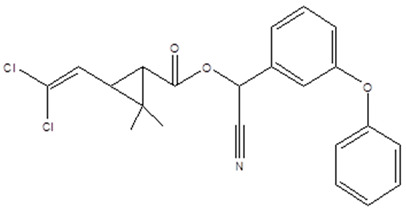 Cypermethrin	C_22_H_19_Cl_2_NO_3_ (415.0.0)	18.21	415.0	163.0
**8**	Bifenthrin	C_23_H_22_ClF_3_O_2_ (422.0)	18.21	424.0 (M^+2^)	181.0
**9**	Cyhalothrin	C_23_H_19_ClF_3_NO_3_ (449.0)	18.21	449.0	181.0
**10**	Dihydroobscurinervinediol diacetate	C_29_H_40_N_2_O_7_ (528.0)	1.63	528.0	69.00
**11**	3,4,5,6-Tetrahydro-6-nonul-2*H*-pyran-2-one (cyclic ester)	C_14_H_25_O_2_ (226.0)	0.70	226.0	99.0
**12**	6-Heptylotetrahydro-2*H*-pyran-2-one (cyclic ester)	C_12_H_22_O_2_ (198.0)	0.44	198.0	99.00
	**Group 4 (Fatty Esters)**				
**1**	Methyl hexadecanoate	C_17_H_34_O_2_ (270.0)	0.28	270.0	74.00
**2**	Methyl octadec-16-enoate	C_19_H_6_O_2_ (296.0)	0.47	296.0	55.00
**3**	Methyl octadec-10-enoate	C_19_H_6_O_2_ (296.0)	0.47	296.0	55.00
**3**	2- (Tetradecycloxyethyl) palmitate	C_32_H_64_O_3_ (496.0)	0.77	496.0	57.00
**4**	Trimyristin	C_45_H_86_O_6_ (722.0)	0.78	722.0	57.00
**5**	(E) -2(stearoyloxy) ethyl octadec-9-enoate	C_38_H_72_O_4_ (592.0)	1.63	592.0	99.00 and 311.0
	**Group 5 (Unsaturated Fatty Esters)**				
**1**	Methyl 5,6-dihydro-5,6-dihydroxy-(5R, 6R)-10′-Apo-α′-PSI-carotenoate	C_28_H_40_O_4_ (440.0)	1.13	440.0	109.0
**2**	Tetrahydrofurfuryl oleate	C_23_H_42_O_3_ (366.0)	1.97	366.0	71.00
**3**	Methyl (10E)-12,12-dideutero-14-oxo-10-nonadecenoate	C_20_H_34_D_2_O_3_ (326.0)	1.63	326.0	99.00
	**Group 6 (Steroides)**				
**1**	17,17-Ethylenedioxy-5,19-cycloaandrast-6-en-3-one	C_21_H_28_O_3_ (328.0)	0.35	328.0	99.00
**2**	(22E)-Ergosta-7,9(11),22-trien-3-yl acetate	C_30_H_46_O_2_ (438.0)	0.35	438.0	43.00
**3**	28-Acetylspirosolan-3-yl acetate	C_31_H_49_NO_4_ (499.0)	0.31	499.0	163.0 and 43.00
**4**	3-Oxo-9 *β*-lanosta-7-en-26,23-olide	C_30_H_46_O_3_ (454.0)	0.42	454.0	439.0
**5**	Cholest-5-en-ol	C_27_H_46_O (386.0)	4.59	386.0	43.00 and 81.00
**6**	3-Methoxy-6-oxo-2′-methylenechloestano [7,8α] cyclobutane	C_31_H_50_O_2_ (454.0)	0.99	454.0	95.00
**7**	17-Acetoxy-4,4-dimethyl-3-methoxy-3,19-epoxy androst-8-en-7-ol	C_24_H_36_O_5_ (404.0)	0.41	404.0	270.0
	**Group 7 (Polyalkene)**				
**1**	2,6,10,15,19,23-Hexa methyl-2, 6,10,14,18,22-tetracosahexaene	C_30_H_5_O (440.0)	0.16	41.0	69.00
	**Group 8 (Heterocyclic)**				
**1**	1-(2-Nitro-4-trifluoro-methyl-phenyl)-5-propyl-1*H*-[1,2,3] triazole-4-carboxylic ethyl ester	C_15_H_15_F_3_N_4_O_4_ (372.0)	3.15	372.0	43.00
**2**	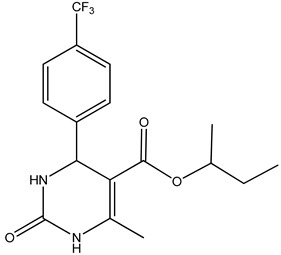 Isobutyl 6-methyl-2-oxo-4-[4-trifluoromethyl)phenyl]-1,2,3,4-tetrahydro-5-pyrimidinearboxylate	C_17_H_19_F_3_N_2_O_3_ (356.0)	9.57	356.0	155.0 and 299.0
	**Group 9 (Aromatic Aldehyde)**				
**1**	m-Phenoxy benzaldehyde	C_13_H_10_O_2_ (198.0)	0.43	198.0	141.0

**Table 2 molecules-26-02635-t002:** Antimicrobial activity of kefir beverage (KB) and neutralized kefir (NKB).

Tested Organism	Inhibition Zone Diameters (mm)		
	KB	NKB	*p*-Value
*Salmonella typhimurium* ATCC14028	17.0 ± 0.5	14 ± 0.2	00.002
*List. monocytogenes* ATCC4957	15 ± 0.2	18 ± 0.0	0.000
*B. cereus* ATCC14579	18 ± 0.45	17 ± 0.18	0.000
*S. aureus* ATCC6538	21 ± 0.44	13 ± 0.25	0.000
*E. coli* ATCC 11229	14 ± 0.2	10 ± 0.1	0.000
*A. flavus* ATCC16872	7 ± 0.18	3 ± 0.0	0.000
*A. niger* ATCC20611	2 ± 0.1	4 ± 0.0	0.000

TCC: American Type Culture Collection.

## Data Availability

The data presented in this study are available in the article.

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
