# Peer review of "Chemical Constitution and Antimicrobial Activity of Kefir Fermented Beverage"

_molecules, 2021, doi:10.3390/molecules26092635_

Round 1

Reviewer 1 Report

The manuscript has been significantly revised and improved. The presentation of the results is now much clearer. It can be published in the "molecules" journal.

Reviewer 2 Report

The authors have corrected most of the issues from the original manuscript. Only correct the word “stereoides” in table 1. It should be “Group 6 (Steroids)”.

This manuscript is a resubmission of an earlier submission. The following is a list of the peer review reports and author responses from that submission.

Round 1

Reviewer 1 Report

The manuscript "Chemical constitution and antimicrobial activity of kefir fermented beverage" by Abdul-Raouf Al-Mohammadi, Rehab Ibrahim, Ahmed H. Mostafa, Ahmed A. Ismaiel, Azza Abou Zeid, Gamal Enan, deals with a topic of great interest that falls within the scope of the journal.  The amount of experimental evidence presented is substantial, but the manuscript is carelessly written, with numerous errors that make it difficult to understand.  In its current state, it is unacceptable for publication in Molecules.   However, the authors are encouraged to resubmit the manuscript after revision with the assistance of a native English speaker.  Please find attached a file with the Abstract and Introduction sections with some suggested corrections.

Reviewer 2 Report

General comment:

The present manuscript deals with the determination of the chemical constitution and the antimicrobial activity of Kefir beverage. This is not novel, and one can find relatively recent reviews in literature dealing with this subject such as CyTA – Journal of Food, 2015, Vol. 13, No. 3, 340–345, (http://dx.doi.org/10.1080/19476337.2014.981588) or Gao & Li, Cogent Food & Agriculture (2016), 2: 1272152 (http://dx.doi.org/10.1080/23311932.2016.1272152). These two papers have not been included in the present manuscript and yet more than 20 self-citations have been included in the References section.

The manuscript is poorly written and presents several English grammar flaws that make the manuscript difficult to read. A thorough revision must be done prior to reconsideration. Regarding the organization, I would suggest the authors to move the “Materials and Methods” section before the “Results” section.

Finally, I will point out some of the most important lines that, in my belief, would be important to be revised. Since there are too many English issues and some other grammatical errors, I will only mention some here

Abstract: correct words such as: “stereoides” (should be: stereoids), “aldhydes” (should be: aldehydes) and expressions such as in line 21: “it was appeared to contain…” Change it for “it seemed to contain…”

Introduction:

  • Lines 28-34: this paragraph is quite hard to understand. It should be rewritten and words typos such as “whih" in line 30 must be corrected. The first 12 citations are self-citations!! I believe that the authors must include other references rather than only their owns.

Results:

  • Line 78: change the line for “…showed about 40 principal peaks which correspond to more than 34 bioactive compounds…”
  • Lines 82-104: there are several errors here. Many chemical names are wrong and do not follow the IUPAC rules. In addition, there are many typos. The authors must change “cm-1” to “cm-1” when refereeing to the IR signals.
  • Line 116: typo in “aetobacter”, it is “acetobacter”.
  • Line 121: here, the word NKB appears for the very first time. The authors should include the meaning. It only appears in the Table 2 title.

Discussion:

  • Lines 201 – 202: the authors discuss about the presence of alcohols in the kefir sample here. According to table 1, there is no alcohol here but rather a tosylate. Correct this.
  • Lines 223-224: there is no need to explain what heterocyclic compounds are as defined by IUPAC. This is well known already.

References: reference 37 has many typing errors.

Figures and Tables:

Figures of molecules in Table 1 must be redrawn properly. The quality of the pictures is quite poor. Please consider using a proper software such as ChemDraw or similar and present all the molecules with the same settings. Also, revise the nomenclature of the compounds. I would suggest the authors to change “(m/e)” to “(m/z)”. Revise typos such as “stereoides”.

In my opinion, figures 3 to 6 are too large. I would suggest combining them in a single figure or take them to a supplementary material file if possible.